# Chemical Composition of Essential Oils from Leaves and Fruits of *Juniperus foetidissima* and Their Attractancy and Toxicity to Two Economically Important Tephritid Fruit Fly Species, *Ceratitis capitata* and *Anastrepha suspensa*

**DOI:** 10.3390/molecules26247504

**Published:** 2021-12-11

**Authors:** Mehmet Kurtca, Ibrahim Tumen, Hasan Keskin, Nurhayat Tabanca, Xiangbing Yang, Betul Demirci, Paul E. Kendra

**Affiliations:** 1Department of Chemistry, Faculty of Science, Selcuk University, 42130 Konya, Turkey; mehmet.kurtca@selcuk.edu.tr; 2Faculty of Health Sciences, Bandirma Onyedi Eylul University, 10200 Bandirma, Turkey; 3Department of Forest Products Chemistry, Faculty of Forestry, Bartin University, 74100 Bartin, Turkey; hkeskin@bartin.edu.tr; 4United States Department of Agriculture-Agricultural Research Service (USDA-ARS), Subtropical Horticulture Research Station (SHRS), 13601 Old Cutler Rd., Miami, FL 33158, USA; nurhayat.tabanca@usda.gov (N.T.); xiangbing.yang@usda.gov (X.Y.); 5Department of Pharmacognosy, Faculty of Pharmacy, Anadolu University, 26470 Eskisehir, Turkey; betuldemirci@gmail.com

**Keywords:** juniper oil, Cupressaceae, *α*-pinene, *α*-thujone, *β*-thujone, cedrol, Mediterranean fruit fly, medfly, Caribbean fruit fly

## Abstract

The present study analyzed the chemical composition of *Juniperus foetidissima* Willd. essential oils (EOs) and evaluated their attractancy and toxicity to two agriculturally important tephritid fruit flies. The composition of hydrodistilled EOs obtained from leaves (JFLEO) and fruits (JFFEO) of *J. foetidissima* was analyzed by GC–FID and GC–MS. The main compounds were *α*-pinene (45%) and cedrol (18%) in the JFLEO and *α*-pinene (42%), *α*-thujone (12%), and *β*-thujone (25%) in the JFFEO. In behavioral bioassays of the male Mediterranean fruit fly, *Ceratitis capitata* (Wiedemann), both JFLEO and JFFEO showed strong attraction comparable to that observed with two positive controls, *Melaleuca alternifolia* and *Tetradenia riparia* EOs. In topical bioassays of the female Caribbean fruit fly, *Anastrepha suspensa* (Loew), the toxicity of JFFEO was two-fold higher than that of JFLEO, with the LD_50_ values being 10.46 and 22.07 µg/µL, respectively. This could be due to differences in chemical components between JFLEO and JFFEO. The JFFEO was dominated by 48% monoterpene hydrocarbons (MH) and 46% oxygenated monoterpenes (OM), while JFLEO consisted of 57% MH, 18% OM, and 20% oxygenated sesquiterpenes (OS). This is the first study to evaluate the attractancy and toxicity of *J. foetidissima* EOs to tephritid fruit flies. Our results indicate that JFFEO has the potential for application to the management of pest tephritid species, and further investigation is warranted.

## 1. Introduction

Resistance to chemical pesticides and the negative effects of chemical residues on the environment, non-target organisms, and human health have led to an urgent need for more environmentally sound strategies for pest management [1,2,3]. Essential oils (EOs) and their rich source of bioactive compounds are considered one of the best resources for the development of alternative insect control agents [1,2,4,5]. The Mediterranean fruit fly or medfly, *Ceratitis capitata* (Wiedemann) (Diptera: Tephritidae) is one of the most destructive and economically important agricultural pests worldwide. The Caribbean fruit fly, *Anastrepha suspensa* (Loew), is found in Cuba, Jamaica, Hispaniola, Puerto Rico, and Florida, USA, where it is a quarantine pest of citrus and a production pest of many specialty fruits, particularly guava. Developing new fruit fly attractants (e.g., host-based kairomones) and alternative plant-based toxicants are promising approaches for improving monitoring tools and pest suppression strategies while minimizing pest management’s environmental impact [6,7]. Thus, the evaluation of EOs for novel bioactive compounds is critical to developing alternatives to conventional pest control methods.

Juniper EOs are well documented for their insecticidal and acaricidal activity against a variety of species, including coleopterans (*Callosobruchus maculatus*, *Sitophilus zeamais*, *S. oryzae*, *Rhyzopertha dominica*, and *Tribolium castaneum*) [8,9,10,11], lepidopterans (*Mythimna separata*, *Plutella xylostella*, and *Spodoptera exigua*) [12,13], a hemipteran (*Brevicoryne brassicae*) [14], dipterans (*Culex pipiens*, *C. quinquefasciatus*, *Aedes aegypti and Anopheles stephensi* [15,16,17,18], and mites (*Tetranychus urticae*) and *Hyalomma aegyptium* [19]. 

Species within the genus *Juniperus* L. (Cupressaceae) are ever-green trees that play an important role in ecology and economy worldwide [20], and juniper derivatives are widely used in the food, cosmetic and spirits industries [21,22]. *Juniperus* extracts have been traditionally used in treatment of tuberculosis, jaundice, hemorrhoids, bronchitis, urinary infection, urticaria, rheumatic arthritis, dysentery and the common cold [23,24,25,26]. Approximately six percent of the medicinal plants in the world are found in Turkey and the country has an important place in terms of natural wealth. Many of these medicinal and aromatic plants are suitable for essential oil production, including *Juniperus foetidissima* Willd [27]. The Turkish Ministry of Agriculture and Forestry reported that medicinal and aromatic plants used in food processing, pharmaceuticals, and the cosmetic industry had an export value of 404 million dollars in 2021 [28]. *Juniperus* is represented by 10 taxa among 7 species, which are mostly distributed in the western, southern, and central parts of Turkey [8]*. Juniperus foetidissima*, locally known as “Kokulu Ardıç” (fragrant juniper), has the reputation for superior grade furniture and wood products due to its fine odor and resistance to many insect pests [20,29,30,31]. 

The current study was conducted to: (i) isolate *J. foetidissima* EOs from the leaves (JFLEO) and fruits (JFFEO) of populations growing wild in central Turkey, (ii) identify and quantify the chemical contents of JFLEO and JFFEO by gas chromatography–flame ionization detection (GC–FID) and gas chromatography–mass spectrometry (GC–MS), (iii) evaluate these oils for their potential attraction of male *C. capitata*, and (iv) determine the toxicities of these oils to the female *A. suspensa*.

## 2. Results and Discussion

In the present study, the yields of JFLEO and JFFEO were 2.36% and 2.98% (v/w), respectively. The range of the yield for *J. foetidissima* EOs reported in other published studies is given in Table 1. Our results for JFLEO yield agree with those reported by Adams [32]. However, our yield of JFFEO was higher than that obtained in all other studies. The high percentage yield in our isolations may be attributed to the favorable climate and soil structure of the region selected for the sample collection. 

### 2.1. GC–MS Analysis 

The chemical composition of the essential oils from *J. foetidissima* leaves and fruits is shown in Table 2. In total, 62 compounds were detected. In the fruit, 44 components were identified, while 60 components were identified in the leaves. The main compounds detected in JFLEO were *α*-pinene (45.2%) and cedrol (18.2%). The major compounds detected in JFFEO were *α*-Pinene (41.9%), *α*-thujone (12.2%) and *β*-thujone (25.1%). (Total ion chromatograms are given in the Appendix A).

Quantitatively, the predominant components of juniper essential oils consisted of monoterpene hydrocarbons (MH; JFLEO, 57%; JFFEO, 48.2%) and oxygenated monoterpenes (OM; JFLEO, 18.1%; JFFEO, 46.3%). Oxygenated sesquiterpenes were higher in leaves as compared to the fruit (OS; JFLEO, 20%; JFFEO, 2.3%) and sesquiterpene hydrocarbons were low in leaves and detected only in trace amounts in the fruit (SH; JFLEO, 3%; JFFEO, 0%) (Figure 1; Table 2).

Although *J. foetidissima* EOs have been investigated in different countries and regions of the same country, they continue to be of interest for both industrial and research purposes. Table 3 summarizes data from literature reports on the main components identified in *J. foetidissima* EOs obtained from the leaves, fruit, branches, and wood [33,34,36,37,46,47,48,49,50]. The chemical composition of JFFEO from the Arasbaran forest in Iran showed significant quantitative and qualitative differences [36,41,47]; sabinene, *α*-pinene, and limonene were the main components, but all three were present in different percentages. In contrast, limonene, *α*-thujone, *β*-thujone and terpinen-4-ol were determined to be the main components in the study of Parvin Salehi et al. [41]. In the study of Lesjak et al. [21] in Macedonia, sabinene and terpinen-4-ol in the leaf oil and only sabinene in the fruit oil were found to be the main components. In another study on JFFEO, Adam collected fruit samples from Greece and reported sabinene, *α*-thujone, and terpinen-4-ol to be the main components [46], whereas samples collected at three locations in Turkey had significantly different components. In samples collected from Amasya, a city in northern Turkey, it was reported that the main components of JFLEO were *α*-pinene and cedrol and the main component of JFLEO was *α*-pinene [34].

Samples from Eskisehir Province (central Anatolia, Turkey) studied showed that cedrol and *β*-thujone were the main components in JFLEO, and that sabinene and limonene were the main components in JFFEO [33]. Another study of samples from Antalya Province, Turkey showed that thujopsene, widdrol, cedrol, and caryophyllene alcohol were the main components in sapwood and heartwood oils [48].

In the present study, the main components were determined to be *α*-pinene and cedrol in JFLEO and *α*-pinene, *α*-thujone, and *β*-thujone in JFFEO; these results agree with those reported by Yaglioglu et al. [34]. Moreover, it is noteworthy that *α*-pinene is the main component in only Turkish and Iranian essential oils. The studies by Tunalier et al. [33] and Yaglioglu et al. [34] were also similar regarding the relatively high content of cedrol. As a general result of these studies, it can be concluded that the chemical composition of volatile compounds in JFEOs may be highly variable, depending on the age of the tree, location, geographic variations, and climatic and ecological conditions of the area in which the tree is located. Considering all these influences, the oil’s composition is highly dependent on these factors. As a result, the JFLEO and JFFEO profiles and the EO yields were quite different and may be of interest to the EO industry utilizing juniper leaf and fruit EOs. 

### 2.2. Biological Activity of Juniperus foetidissima Essential Oils

#### 2.2.1. Attraction of the Mediterranean Fruit Fly

In short-range attraction bioassays of male *C. capitata*, there were significant differences in their behavioral response to the six essential oil treatments (*F* = 7.107; df = 5.24; *p* < 0.001; Figure 2). Both JFLEO and JFFEO were found to be highly attractive to males, with results comparable to those observed in the two positive controls, tea tree essential oil (TTO) [49] and *Tetradenia riparia* essential oil (TREO) [50]. Their attraction to the best three oils—JFFEO, TTO, and TREO—was significantly greater than that observed with blue tansy essential oil (BTEO) [51] or mastic gum essential oil (MGEO) [52]. 

Comparing the two JFEOs, the higher response to JFFEO suggests that the activity may be related to the thujone concentration. JFFEO contains a higher percentage of both *α*- and *β*-thujone (12.2% and 25.1%, respectively), which accounted for 37% of the total composition (Table 2). The isomers *α*- and *β*-thujone are monoterpene ketones and naturally occur in the essential oils of *Thuja occidentalis*, *Tanacetum vulgare*, *Salvia officinalis*, and *Artemisia absinthium* [53]. Previous studies highlighted that the thujone accumulation reaches higher levels in the leaf and budding floral stages and changes after the flowering and during the seed ripening in *S. officinalis* and *A. absinthium* EOs [53]. In the case of *T. vulgare*, *β*-thujone in the flower was higher than it was in the leaf oil in a Dutch tansy oil [54]. However, literature data reveals that the thujone content can vary depending on the plant species, harvested time, and region or habitat from which the sample originates [53]. Comparing previously reported *Juniperus* EOs, thujones were detected in relatively low concentrations or were lacking in nine *Juniperus* EOs (*J. chinensis*, *J. communis*, *J. excelsa*, *J. macropoda*, *J.*
*phoenicea*, *J. saltuaria*, *J. scopulorum*, *J. squamata*, and *J. virginiana)*, whereas thujones were found in higher amounts in *J. foetidissima EOs* [31,33,53,55,56,57,58]. 

Numerous studies reported that thujones exhibit various types of bioactivity, such as antifeedant activity against Sitka black-tailed deer (*Odocoileus hemionus sitkensis*) [59], toxicity to the larvae and adults of the sycamore lace bug (*Corythucha ciliata*) [60], high toxicity to western corn rootworm larvae (*Diabrotica virgifera virgifera*), and less toxicity to the two-spotted spider mite (*Tetranychus urticae*) and housefly (*Musca domestica*) [61]. In choice and no-choice tests with the peach-potato aphid (*Myzus persicae*), as well as in electropenetrography (EPG) analyses, *β*-thujone caused changes in the aphid’s feeding behavior, while *α*-thujene did not evoke any changes [62]. The authors conducted further studies on *β*-thujone derivatives, and found that *β*-thujone and four *β*-thujone derivatives showed an antifeedant activity against *M. persicae*, which might be used for selective integrated pest management (IPM) control strategies against peach-potato aphids. 

Our present results identify JFFEO as an effective new attractant to male *C. capitata* with the potential to improve field lures for this pest. The major constituents detected in JFFEO were *α*-pinene (41.9%), *β*-thujone (25.1%), and *α*-thujone (12.2%). These compounds are different than those identified previously in other EOs highly attractive to the medfly. The predominant components in TTO include terpinen-4-ol (41.8%), γ-terpinene (15.5%), and *p*-cymene (11.9%) [49]. The primary components in TREO consist of fenchone (15%), δ-cadinene (11%), 14-hydroxy-*β*-caryophyllene (8%), and τ-cadinol (7%) [50]. In addition, early work on Angelica seed oil identified *α*-copaene as a strong attractant to male *C. capitata* [63]. Further studies are needed to determine whether combinations of two or more of these EOs result in an increase in their attraction of medflies. For other pest insects, improved lures have been developed through a combination of multiple kairomones that work synergistically [64]. 

#### 2.2.2. Toxicity to the Caribbean Fruit Fly

The present study demonstrated that *J. foetidissima* EOs had strong toxicity to adult female *A. suspensa*. The median lethal doses (LD_50_) of JFLEO and JFFEO for *A. suspensa* were 22.07 (χ^2^ = 7.232, df = 4, *p* = 0.1241) and 10.45 µg/µL (χ^2^ = 6.6917, df = 4, *p* = 0.1531), respectively (Table 4). Our results also showed that an untreated control had a 0% adult mortality during the test, and a topical application with acetone alone resulted in a 6.7% adult mortality. A dose of 10.46 µg/µL of JFFEO was required to achieve a 50% mortality of *A. suspensa*, which was less than half the dose of JFLEO required, 22.07 µg/µL. This could be due to the different chemical compositions of the two essential oils. The primary component of both *J. foetidissima* EOs is *α*-pinene, which has demonstrated insecticidal effects on several insect pests including the medically important head louse, Pediculus humanus, and a stored product pest, the maize weevil, *Sitophilus zeamais* [65,66]. In *S. zeamais*, *α*-pinene also demonstrated a reduction in their progeny of up to 98% at a dose of 12 ppm via contact. In addition, monoterpenes such as *α*-pinene are typically volatile chemicals with a demonstrated fumigation toxicity to some insect pests, including *S. zeamais.* [66]. Our results showed that JFFEO had a stronger toxicity to *A. suspensa* female adults than JFLEO. This could be due to other chemicals in the composition (e.g., *α*-thujone and *β*-thujone) of JFFEO that may contribute to synergistic insecticidal effects on *A. suspensa*. For example, (*α* + *β*) thujone showed toxicity to the red imported fire ant (*Solenopsis invicta*) when used as a fumigant and achieved a 100% mortality with an appropriate dose-dependent treatment [67]. It is speculated that a synergistic effect between *α*-pinene and (*α* + *β*) thujone may have played an important role in improving the toxicity. However, further studies are needed to verify the synergistic effect. Our results demonstrated a strong contact toxicity of the two EOs to *A. suspensa*; they may also have the potential to work as fumigants in insect pest control due to their volatile attributes. Therefore, *J. foetidissima* EOs may improve fruit fly pest control programs through a combination of both contact and fumigation treatments, but further studies are needed to justify the feasibility.

## 3. Materials and Methods

### 3.1. Plant Material

Leaves and fruits of *J. foetidissima* were collected from Beypazari town in Ankara Province, Turkey (Appendix A). The specimens were authenticated by Dr. Barbaros Yaman and have been deposited in the Herbarium of the Faculty of Forestry, Bartin University (BOF 516). Fresh and undamaged samples were collected and stored at −24 °C until the laboratory analyses. 

### 3.2. Essential Oil Isolation

Essential oils of the leaf and fruit of *J. foetidissima* were obtained by hydrodistillation using a Clevenger apparatus (Ildam Cam Ltd., Ankara, Turkey). One hundred grams each of fresh leaves and fruits were used, and the oils were collected for 3 h [68,69]. The samples were dried with anhydrous sodium sulphate in a sealed vial until analyses [70]. 

### 3.3. GC-MS Analysis

The GC–MS analysis was carried out with an Agilent 5975 GC–MSD system (Agilent Technologies, Santa Clara, CA, USA). An Innowax FSC column (60 m × 0.25 mm, 0.25 μm film thickness) was used with helium as the carrier gas (0.8 mL/min). GC oven temperature was kept at 60 °C for 10 min, programmed to 220 °C at a rate of 4 °C/min and kept constant at 220 °C for 10 min, and then programmed to 240 °C at a rate of 1 °C/min. The split ratio was set at 40:1. The injector temperature was set at 250 °C. Mass spectra were recorded at 70 eV. Mass range was from *m/z* 35 to 450.

### 3.4. GC Analysis 

The GC analysis was carried out using an Agilent 6890N GC system (Agilent Technologies, Santa Clara, CA, USA). FID detector temperature was 300 °C. To obtain the same elution order as with GC–MS, simultaneous auto-injection was done on a duplicate of the same column applying the same operational conditions. Relative percentage amounts of the separated compounds were calculated from FID chromatograms. The analysis results are given in Table 2.

### 3.5. Identification of the Compounds

Compounds were identified by comparison of the chromatographic peaks retention times with those of standards analyzed under the same conditions, and by comparison of the retention indices (RI), as Kovats indices, [71] as well as MS literature data [72,73]. Comparisons of MS fragmentation patterns with those of standards and mass spectrum database search were performed using the commercial Wiley GC–MS Library, MassFinder Software 4.0 [74] and the in-house “Baser Library of Essential Oil Constituents” built up by genuine compounds and components of known oils.

### 3.6. Laboratory Bioassays

#### 3.6.1. Short-Range Attraction Bioassays with *Ceratitis capitata*

Sterile male *C. capitata* were used in all bioassays. The source of insects, laboratory rearing procedures, and short-range attraction bioassays were identical to those described previously [49]. All tests were conducted at room temperature in screened cages (20.3 × 20.3 × 20.3 cm) into which 50 flies were introduced 1 h prior to initiation of an experiment. Assays were started by introducing a Petri dish (53 mm diameter × 12 mm height) containing a filter paper disk (Whatman #1, 3.5 cm diameter) soaked with the test EO (10 µL of a 10% dilution in acetone). Separate cages were used for each test to observe medfly response to: (1) JFLEO, *Juniperus foetidissima* leaf essential oil; (2) JFFEO, *J. foetidissima* fruit essential oil; (3) TTO, tea tree oil, a known strong attractant [49] derived from *Melaleuca alternifolia* (Maiden and Betche) Cheel (Essential Oil India–SAT Group, Kannauj, India); (4) TREO, *Tetradenia riparia* essential oil, a known strong attractant obtained from *Tetradenia riparia* (Hochst.) Codd as described in Blythe et al. [50]; (5) BTEO, blue tansy essential oil, a mild attractant obtained from *Tanacetum anuum* L. as described in Stappen et al. [51]; and (6) MGEO, mastic gum essential oil, a mild attractant obtained from *Pistacia lentiscus* L. var. *chia* as described in Tabanca et al. [52]. After 30 min, response was recorded as the number of flies within a Petri dish, which was subsequently converted to the percentage of flies attracted. All tests were replicated five times, and the position of the cages was randomized between replicate runs. 

#### 3.6.2. Toxicity of JFEOs to *Anastrepha suspensa*

Topical bioassays using thoracic application to adult female *Anastrepha suspensa* were conducted to determine the toxicities of JFLEO or JFFEO under laboratory conditions at 26 ± 1 °C, 70 ± 5% RH, and 12:12 L:D photoperiod. To prepare the stock solution, 100 µg of JFLEO or JFFEO was diluted in 1 µL of dimethyl sulfoxide (DMSO) to establish a 100 µg/µL solution. To evaluate the toxicities of the two EOs, a serial dilution of stock solution was prepared with acetone to establish 1.25, 2.5, 5, 10, 15, and 50 µg/μL solutions, and each dilution was tested in topical bioassays.

To conduct topical bioassays, pupae of *A. suspensa* were collected in a tray and placed inside a screen cage (30 cm × 30 cm × 30 cm) under the laboratory conditions described above to allow for adult emergence. Newly emerged female adults (<3 d old) were collected using an aspirator into a plastic vial (3 cm in diameter × 8 cm in height). Female adults in the vial were first chilled at 4 °C in a refrigerator for 5 min to calm the flies, and calmed flies were then removed from the refrigerator to a petri dish to facilitate the topical application. A repeating dispenser equipped with gastight and microliter syringe (50 µL) (PB600, Hamilton Company, Reno, NV, USA) was used to apply 1 µL dilution at each concentration of each EO on the dorsal thorax of the calmed adult flies. After topical application, the adult flies were immediately transferred into a plastic cup (6 cm in diameter × 7.4 cm in height) and covered with a mesh screen for post-treatment observation. After 24 h, numbers of live and dead flies were documented and mortality of *A. suspensa* in each treatment was calculated. Untreated female adults and those treated with acetone alone were used as controls. For each dilution, 10 female adult flies were treated, and each treatment was replicated 3 times. In total, 240 female flies were used in the evaluation of each EO.

## 4. Statistical Analysis

For the attraction bioassays with *C. capitata*, an analysis of variance (ANOVA) followed by a mean separation with Tukey’s HSD test (*p* < 0.05) was used to analyze results (SigmaPlot version 14, Systat Software, Inc.: San Jose, CA, USA) [75]. 

Mortality data of *A. suspensa* in toxicity bioassays were used to calculate the median lethal dose (LD_50_) for each EO (JFLEO or JFFEO). Mortality data for each treatment were corrected by mortalities in the untreated control using Abbott’s formula [76] prior to the analysis. A probit analysis was then used to calculate the lethal dose corresponding to a 50% reduction (LD_50_) in the *A. suspensa*’s survival based on the regression curve. The statistical analysis was performed using SAS version 9.4 [77].

## 5. Conclusions

In this study, the chemical composition of essential oils from the *J. foetidissima* leaf (JFLEO) and fruit (JFFEO) were investigated. Our study is the first report to evaluate the attractancy and toxicity of *J. foetidissima* oils to *C. capitata* and *A. suspensa*. The major compounds identified in JFLEO were *α*-pinene and cedrol, while JFFEO contained *α*-pinene, *α*-thujone, and *β*-thujone. The differences in chemical composition between the two EOs may account for the differences in their attractiveness and toxicity observed in two tephritid species. JFFEO was found to be more effective than JFLEO in attracting male *C. capitata*, which might be attributed to the *α-* and *β*-thujone content of JFFEO. Furthermore, JFFEO also showed a much higher toxicity to female *A. suspensa* than JFLEO. Future studies are needed to separate the diastereomers of thujone and evaluate their chemical activities on tephritid species. Further evaluation of plant EOs as novel attractants and toxicants is highly desirable, as they are biodegradable and therefore environmentally friendly. This current report concludes that JFLEO and JFFEO show promise for integration into management programs for major fruit fly pests. 

## Figures and Tables

**Figure 1 molecules-26-07504-f001:**
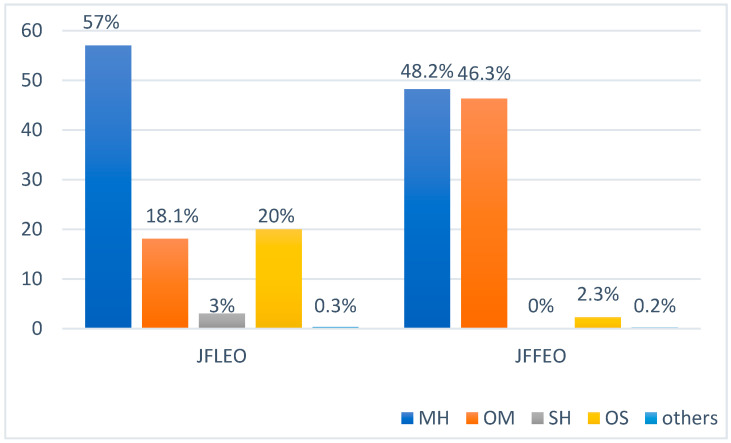
Percent composition of terpene groups in the EOs from the leaves and fruit of *J. foetidissima*. MH—monoterpene hydrocarbons; OM—oxygenated monoterpenes; SH—sesquiterpene hydrocarbons; OS—oxygenated sesquiterpenes.

**Figure 2 molecules-26-07504-f002:**
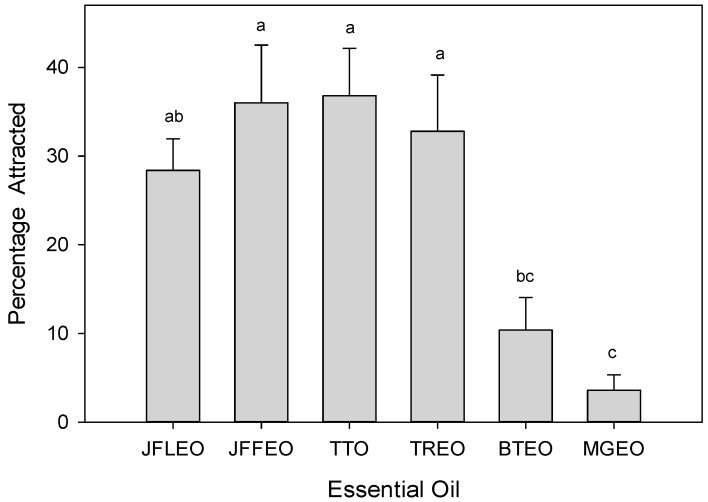
*Juniperus foetidissima* leaf essential oil (JFLEO) and fruit essential oil (JFFEO) vs. controls: two strong attractants (TTO, TREO) and two mild attractants (BTEO, MGEO). Bars topped with the same letter are not significantly different (Tukey’s HSD mean separation; *p* < 0.05).

**Table 1 molecules-26-07504-t001:** Comparison of oil yields of JFLEO and JFFEO reported in the literature.

Part of *J. foetidissima*	%	References
Leaves (needles)	0.30	[33]
	1.03	[34]
	1.40	[35]
	2.26	[32]
	3.00	[36]
	0.68	[37]
	0.60	[38]
	0.50	[39]
	1.38	[40]
Fruits	0.90	[33]
	0.64	[34]
	2.03	[37]
	1.70	[41]
	2.30	[38]
	0.30	[39]

**Table 2 molecules-26-07504-t002:** Composition of the essential oils from leaves (JFLEO) and fruit (JFFEO) of *Juniperus foetidissima*.

	Nr	KI ^a^	RRI ^b^	Compound	JFLEO %	JFFEO %	IM
Monoterpene hydrocarbons (MH)	1	1012 ^c^	1014	Tricyclene	0.2	0.2	MS
	2	1025 ^d^	1032	*α*-pinene	45.2	41.9	RRI, MS
	3	1026 ^c^	1035	*α*-thujene	0.5	0.2	MS
	4	1061 ^c^	1072	*α*-fenchene	0.3	tr	MS
	5	1077 ^e^	1076	Camphene	0.4	0.1	RRI, MS
	6	1117 ^d^	1118	*β*-pinene	0.5	0.7	RRI, MS
	7	1122 ^c^	1132	Sabinene	1.1	1.2	MS
	8	1122 ^c^	1135	Thuja-2,4(10)-diene	0.1	0.1	MS
	9	1122–1169 ^c^	1159	δ-3-carene	1.1	0.2	MS
	10	1160 ^c^	1174	Myrcene	0.8	0.9	RRI, MS
	11	1248 ^c^	1187	*o*-cymene	0.1	tr	MS
	12	1212 ^e^	1203	Limonene	3.0	1.0	RRI, MS
	13	1188–1233 ^c^	1218	*β*-phellandrene	0.1	tr	MS
	14	1282 ^e^	1280	*p*-cymene	3.6	1.7	RRI, MS
Oxygenated monoterpenes (OM)	15	1331–1384 ^c^	1384	*α*-pinene oxide	0.5	1.4	MS
	16	1399 ^c^	1406	*α*-fenchone	tr	-	RRI, MS
	17	1423 ^c^	1437	*α*-thujone	0.2	12.2	MS
	18	1439 ^c^	1451	*β*-thujone	6.7	25.1	MS
	19		1458	*cis*-1,2-limonene epoxide	0.7	-	MS
	20	1459 ^e^	1474	*trans*-sabinene hydrate	-	0.2	MS
	21	1473 ^e^	1481	Fencyl acetate	0.1	-	MS
	22	1486 ^f^	1499	*α*-campholene aldehyde	0.1	0.1	MS
	23	1515 ^c^	1532	Camphor	tr	tr	RRI, MS
	24	1543 ^c^	1553	Linalool	0.1	-	RRI, MS
	25		1556	*cis*-sabinene hydrate	-	0.1	MS
	26	1584 ^c^	1571	*trans-p*-Menth-2-en-1-ol	0.1	0.1	MS
	27	1575 ^c^	1586	Pinocarvone	-	0.1	RRI, MS
	28	1568 ^f^	1591	Fenchyl alcohol	0.7	-	RRI, MS
	29	1579 ^c^	1591	Bornyl acetate	1.2	0.5	RRI, MS
	30	1601 ^c^	1611	Terpinen-4-ol	3.1	2.7	RRI, MS
	31	1614 ^c^	1638	*cis-p*-menth-2-en-1-ol	0.2	0.1	MS
	32	1631 ^c^	1648	Myrtenal	0.3	0.3	MS
	33		1658	Sabinyl acetate	0.8	0.9	MS
	34	1661 ^c^	1670	*trans*-pinocarveol	0.3	0.3	RRI, MS
	35	1680 ^c^	1683	*trans*-verbenol	0.9	0.9	RRI, MS
	36	1694 ^c^	1706	*α*-terpineol	0.7	0.3	RRI, MS
	37	1699 ^c^	1719	Borneol	0.2	0.1	RRI, MS
	38	1720 ^c^	1725	Verbenone	0.4	0.4	RRI, MS
	39	1743–1808 ^c^	1804	Myrtenol	0.2	0.2	MS
	40	1836 ^c^	1845	*trans*-carveol	0.3	0.1	RRI, MS
	41	1813–1865 ^c^	1864	*p*-Cymen-8-ol	0.3	0.2	RRI, MS
Sesquiterpene hydrocarbons (SH)	42		1519	1,7-diepi-*α*-cedrene (=*α*-funebrene)	0.1	tr	MS
	43	1563–1608 ^c^	1577	*α*-cedrene	0.6	-	MS
	44		1594	1,7-diepi-*β*-cedrene (=*β*-funebrene)	0.9	tr	MS
	45	1574–1647 ^c^	1613	*β*-cedrene	0.5	-	MS
	46	1597–1648 ^c^	1644	Widdrene (*=Thujopsene*)	0.2	-	MS
	47	1649 ^c^	1661	Alloaromadendrene	0.4	-	MS
	48		1747	*α*-alaskene	0.2	-	MS
	49	1766–1849 ^c^	1849	Cuparene	0.2	tr	MS
	50	2146–2256 ^c^	2256	Cadalene	0.1	-	MS
Oxygenated sesquiterpenes (OS)	51	2088 ^c^	2088	1-*epi*-cubenol	0.2	-	MS
	52		2100	*allo*-cedrol	1.0	0.2	MS
	53	2093–2149 ^c^	2143	cedrol	18.2	2.1	RRI, MS
	54		2170	*epi*-cedrol	0.2	-	MS
	55	2165 ^e^	2187	τ-cadinol	0.1	-	MS
	56	2135–2219 ^c^	2219	δ-cadinol (=*α*-muurolol)	0.1	-	MS
	57		2255	*α*-cadinol	0.2	-	MS
Others	58	1056–1106^c^	1093	Hexanal	tr	0.2	RRI, MS
	59		1379	3-methyl-3-butenyl isovalerate	0.1	-	MS
	60		1452	*α*,*p*-dimethylstyrene	0.1	tr	MS
	61		1617	Hexyl hexanoate	0.1	-	RRI, MS
	62		1797	*p*-methyl acetophenone	tr	tr	MS
				**Total**	98.6	97.0	

^a^ KI from the literature; ^c,d,e,f^ [42,43,44,45]; ^b^ RRI—relative retention indices calculated against *n*-alkanes, % calculated from FID data; tr—trace (<0.1%); IM—the identification method based on the relative retention indices (RRI) of authentic compounds on the HP Innowax column and by matching mass spectra (MS) with those of the Wiley and MassFinder libraries and comparisons with literature data.

**Table 3 molecules-26-07504-t003:** Predominant components of EOs isolated from different parts of *J. foetidissima*, as reported in the literature [33,34,36,37,46,47,48,49,50].

Part of *J. foetidissima*	Main Components	%	References
Leaves (needles)	sabinene	39.9	[37]
		15.9	[46]
		19.9	[36]
	cedrol	11.4	[33]
		25.5	[34]
	α-pinene	56.1	[34]
		22.2	[36]
	α-thujone	18.6	[46]
	β-thujone	26.5	[33]
	terpinen-4-ol	17.0	[37]
		17.6	[46]
	limonene	20.9	[36]
Fruit	sabinene	23.7	[33]
		29.9	[37]
		27.1	[47]
		37.1	[36]
	α-pinene	90.2	[34]
		19.8	[47]
		29.9	[36]
	limonene	13.1	[33]
		25.5	[47]
		11.8	[36]
		22.29	[41]
	α-thujone	13.46	[41]
	β-thujone	22.32	[41]
	terpinen-4-ol	15.4	[41]
Branches	sabinene	19.1 (male) 34.3 (female)	[47]
	α-pinene	17.4 (male) 8.7 (female)	
	limonene	36.3 (male) 5.8 (female)	
Wood	thujopsene	19.82 (sapwood) 23.78 (heartwood)	[48]
	widdrol	3.0 (sapwood) 15.83 (heartwood)	
	cedrol	18.8 (sapwood) 10.0 (heartwood)	
	caryophyllene alcohol	25.0 (sapwood) 8.9 (heartwood)	

**Table 4 molecules-26-07504-t004:** Median lethal dose (LD_50_) of *J. foetidissima* EO for the control of the adult female Caribbean fruit fly, *A. suspensa*, under laboratory conditions.

*J. foetidissima* EO	*n*	Slope (±SE)	LD_50_ (95% FL)	χ^2^	df	*p*
JFLEO	240	1.34 ± 0.12	22.07 (17.56–29.34)	7.2320	4	0.1241
JFFEO	240	2.10 ± 0.15	10.45 (9.13–12.08)	6.6917	4	0.1531

A probit analysis was used to calculate the LD_50_ lethal dose of *J. foetidissima* oils in adult females of *A. suspensa* (PROC PROBIT, SAS Institute Inc.: Cary, NC, USA, 2020).

## Data Availability

Data is contained within the article.

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
