# Peer review of "Chemical Composition of Essential Oils from Leaves and Fruits of Juniperus foetidissima and Their Attractancy and Toxicity to Two Economically Important Tephritid Fruit Fly Species, Ceratitis capitata and Anastrepha suspensa"

_molecules, 2021, doi:10.3390/molecules26247504_

Round 1

Reviewer 1 Report

Please find my comments as follows;

  1. The title of the article is misleading; the authors mentioned the title as "Chemical Composition of Juniperus foetidissima Leaf and Fruit Oils and their Attractancy and Toxicity for Two Economically Important Tephritid Fruit Fly Species, Ceratitis capitata and Anastrepha suspensa". However, later in the manuscript the authors discussed about the essential oil from the plant. Authors needs to understand that the oils are generally considered as triglycerides, whereas essential oils are totally different and highly volatile stress response elements.
  2. I think the authors intention is to control the population of tephritid fly population using natural agents. Hence, authors needs to restructure the introduction section as follows. Initially describe the significance/ economic loss associate with the Tephritid Fruit Flies. The authors can then mention the significance of different essential oils in the management of various pests and later come in to the topic of Juniperus foetidissima essential oils. This will provide a proper direction to the article.
  3. Authors mentioned about the chemical composition of the essential oil. However, it is also important to mention the yield of essential oil and also how many different preparations (duplicate/ triplicate) were made on each essential oil. It is always recommended that authors prepare the same essential oil in three independent extractions, compare them for the chemical composition and yield. If all are similar, you can mix the three preparations for future studies. This will enhance the value of your study and also improves the reliability of your data.
  4. Authors may provide the details of literature based on which they carried out the extraction of essential oil. If the conditions are standardised by the authors, it can be mentioned in methodology. Also authors needs to move the yield details in to the results section.
  5. Authors needs to incorporate the chromatogram of the GC-MS analysis as a supplementary material, only if it is of good quality. (If the authors are outsourcing the MS analysis, then it will be difficult to obtain a good quality figure.). It is always difficult to have all the instruments in one department.
  6. What is the intention of authors in keeping the figure 1 in dark mode? I think a plain figure as given in figure 2 may be appropriate.

Author Response

1. The title of the article is misleading; the authors mentioned the title as "Chemical Composition of Juniperus foetidissima Leaf and Fruit Oils and their Attractancy and Toxicity for Two Economically Important Tephritid Fruit Fly Species, Ceratitis capitata and Anastrepha suspensa". However, later in the manuscript the authors discussed about the essential oil from the plant. Authors needs to understand that the oils are generally considered as triglycerides, whereas essential oils are totally different and highly volatile stress response elements

Response 1:

The title changed to “Chemical Composition of Essential Oils from Leaves and Fruits of Juniperus foetidissima   and their Attractancy and Toxicity for Two Economically Important Tephritid Fruit Fly Species, Ceratitis capitata and Anastrepha suspensa

2. I think the authors intention is to control the population of tephritid fly population using natural agents. Hence, authors need to restructure the introduction section as follows. Initially describe the significance/ economic loss associate with the Tephritid Fruit Flies. The authors can then mention the significance of different essential oils in the management of various pests and later come into the topic of Juniperus foetidissima essential oils. This will provide a proper direction to the article.

Response 2:

In the first paragraph, in order, it has been mentioned that there is an urgent need for environmentally friendly strategies due to the effects of chemical pesticides on the environment and human health. After that essential oils (EOs) and their rich bioactive compounds are some of the best sources for the development of alternative insect control agents. It was mentioned that Turkey has rich natural juniper resources and therefore, it was stated that these essential oils should be used especially in this field.

3. Authors mentioned about the chemical composition of the essential oil. However, it is also important to mention the yield of essential oil and also how many different preparations (duplicate/ triplicate) were made on each essential oil. It is always recommended that authors prepare the same essential oil in three independent extractions, compare them for the chemical composition and yield. If all are similar, you can mix the three preparations for future studies. This will enhance the value of your study and also improves the reliability of your data.

Response 3:

Absolutely correct, but distillation was only possible for one time, due to the lack of sufficient plant material. Our future work should be decided on different preparations and to compare them for the chemical composition and yield.

4. Authors may provide the details of literature based on which they carried out the extraction of essential oil. If the conditions are standardized by the authors, it can be mentioned in methodology. Also, authors need to move the yield details in to the results section.

Response 4:

Essential oils of the leaf and fruit of J. foetidissima were obtained by hydrodistillation using a Clevenger type apparatus (Ildam Cam Ltd. Ankara-Turkey) according to the method recommended in the European Pharmacopoeia (European Pharmacopoeia, Council of Europe, 5th ed., 2004: Strasbourg, Vol.1, 2005, p.217). The literatures were added. The EOs yield moved to Results and Discussion. Also, a new table was added related to EOs literature.

5. Authors needs to incorporate the chromatogram of the GC-MS analysis as a supplementary material, only if it is of good quality. (If the authors are outsourcing the MS analysis, then it will be difficult to obtain a good quality figure.). It is always difficult to have all the instruments in one department.

Response 5:

Chromatograms are included.

6. What is the intention of authors in keeping the figure 1 in dark mode? I think a plain figure as given in figure 2 may be appropriate

Response 6:

Figure 1 was corrected.

Reviewer 2 Report

Manuscript ID    molecules-1490032

Research paper entitled "Chemical Composition of Juniperus foetidissima Leaf and Fruit Oils and their Attractancy and Toxicity for Two Economically Important Tephritid Fruit Fly Species, Ceratitis capitata and Anastrepha suspensa", contain good information Juniperus foetidissima leaf and fruit essential oils and their Attractancy and Toxicity. However, there is some recommendation for this research.

  • In the Section:Introduction

Some plants have good properties, but are more expensive to grow and harvest than others, which limits their uses. So you have to discuss the price and availability of Juniperus foetidissima leaf and fruit essential oils, to show the added value of your choice of these essential oils.

  • In the section:2. Essential Oil Isolation

The author cited ‘The yields of Juniperus foetidissima leaf and J Juniperus foetidissima fruit were 2.36 and 2.98% (v/w), respectively’.

     The essential oils of genus Juniperus has been the subject of several studies. For this reason, it is necessary to improve the discussion of essential oil yields of Juniperus foetidissima in comparison with other work, and added this comparison in the results and discussion section.

  • How are the results of this research different from the ones already published in the literature as regards the performance of your plant essential oils?

Author Response

Research paper entitled "Chemical Composition of Juniperus foetidissima Leaf and Fruit Oils and their Attractancy and Toxicity for Two Economically Important Tephritid Fruit Fly Species, Ceratitis capitata and Anastrepha suspensa", contain good information Juniperus foetidissima leaf and fruit essential oils and their Attractancy and Toxicity. However, there is some recommendation for this research.

1. In the Section: Introduction

Some plants have good properties, but are more expensive to grow and harvest than others, which limits their uses. So, you have to discuss the price and availability of Juniperus foetidissima leaf and fruit essential oils, to show the added value of your choice of these essential oils. (Ä°brahim)

Response 1:

We added sentences about medicinal and aromatic plants value include J.foetidissima and the others.

2. In the section:2. Essential Oil Isolation

The author cited ‘The yields of Juniperus foetidissima leaf and J Juniperus foetidissima fruit were 2.36 and 2.98% (v/w), respectively’.

The essential oils of genus Juniperus has been the subject of several studies. For this reason, it is necessary to improve the discussion of essential oil yields of Juniperus foetidissima in comparison with other work, and added this comparison in the results and discussion section.

Response 2:

A new table was added related with EOs literature and our EOs yield compared to this table.

3. How are the results of this research different from the ones already published in the literature as regards the performance of your plant essential oils?

Response 3:

This study represents the first investigation of J. foetidissima essential oils (EOs) from the leaves (JFLEO) and fruits (JFFEO) against tephritid fruit flies. In lab bioassays with male medfly, both JFLEO and JFFEO showed strong attraction, equal to that observed with two positive controls, tea tree (Melaleuca alternifolia), and African ginger bush (Tetradenia riparia) EOs.  In topical bioassays with female Caribbean fruit fly, the toxicity of JFFEO was two-fold greater than JFLEO due to differences in chemical components. The higher response to JFFEO  may be related to the thujone concentration. Insecticidal activity of thujones to other insect species was discussed in the manuscript. The results of this study will be useful for researchers interested in exploring the natural-based management programs for two economically important fruit fly pests.

Reviewer 3 Report

The present study analyzed the composition of Juniperus foetidissima leaves and fruits essential oils (EOs) and evaluated their attractancy and toxicity against adults of Ceratitis capitata and Anastrepha suspensa. The work is well structured, fluent in reading. Therefore, I only suggest minor revisions.

Abstract

Line 21: Change analyzed with analysed and leaf with leaves.
Line 22: Change fruit with fruits.
Line 24: Here, too, change the analyzed to analyzed
Line 29: Change fruit fly with fruit flies.

Introduction

Line 41: Change members with species.
Line 61: Change Mediterranean fruit fly or medfly in Mediterranean fruit flies or medflies.
Line 72: Change conditions in factors.
Line 74: Change fruit in fruits.

Results and discussion

Line 81: Change the opening sentence to The chemical composition of the essential oils of the leaves and fruits of etc..
Line 82: Change fruit in fruits.
Line 87: Modify leaf and fruit to leaves and fruits.
Line 95 and 96: Change fruit to fruits.

Line 109: et without the point.

3.2 Essential oils isolation

This paragraph does not indicate the quantity of fruits and leaves: insert them.

Table 1 presents limits to be overcome. To confirm the single compounds it is necessary either to inject all the single identified metabolites (Impossible!) or to report the RRI value obtained for the oils with the non-polar column. Therefore, it is necessary to enter the values ​​also for the non-polar column and indicate which compounds have been confirmed by comparisons with authentic standards.

Throughout the work, change fruit into fruits and leaf into leaves where necessary!

Author Response

The present study analyzed the composition of Juniperus foetidissima leaves and fruits essential oils (EOs) and evaluated their attractancy and toxicity against adults of Ceratitis capitata and Anastrepha suspensa. The work is well structured, fluent in reading. Therefore, I only suggest minor revisions.

Abstract

Line 21: Change analyzed with analysed and leaf with leaves.

American English uses "analyzed" and we would like to keep it as "analyzed".

Line 22: Change fruit with fruits. Corrected.
Line 24: Here, too, change the analyzed to analyzed. Both are the same.

Line 29: Change fruit fly with fruit flies. Corrected.

Introduction

Line 41: Change members with species. Corrected.
Line 61: Change Mediterranean fruit fly or medfly in Mediterranean fruit flies or medflies. Corrected.
Line 72: Change conditions in factors. Corrected.
Line 74: Change fruit in fruits. Corrected.

Results and discussion

Line 81: Change the opening sentence to the chemical composition of the essential oils of the leaves and fruits of etc. Corrected.

Line 82: Change fruit in fruits. Corrected.
Line 87: Modify leaf and fruit to leaves and fruits. Corrected.
Line 95 and 96: Change fruit to fruits. Corrected.
Line 109: et without the point Corrected.

3.2 Essential oils isolation

1. This paragraph does not indicate the quantity of fruits and leaves: insert them.

Response 1:

The amounts of fruits and leaves were written for the essential oils. One hundred grams of each fresh tissue sample was used to obtain the essential oils according to the method recommended in the European Pharmacopoeia. Literature (European Pharmacopoeia, vol. 1., 5th ed. Council of Europe, Strasbourg, pp. 217) was added.

2. Table 1 presents limits to be overcome. To confirm the single compounds, it is necessary either to inject all the single identified metabolites (Impossible!) or to report the RRI value obtained for the oils with the non-polar column. Therefore, it is necessary to enter the values ​​also for the non-polar column and indicate which compounds have been confirmed by comparisons with authentic standards.

Response 2:

Analyses were performed on a polar column. In the revised table, literature RRI values for the polar column were added. We already indicated in the table, which compounds were confirmed by comparisons with authentic standards, as seen in the Identification method column. If we identify compounds with authentic standards, we state them as RRI and MS.

Throughout the work, change fruit into fruits and leaf into leaves where necessary! Corrected.

Round 2

Reviewer 1 Report

I think authors have took their time and made significant revisions in their manuscript. I think there only require minor corrections in terms of grammar or punctuation, which can be done during proof correction. Hence, inaccept the revised manuscript for publication in the journal.